# Predictive Factors and Oncologic Outcome of Downgrade to Pathologic Gleason Score 6–7 after Radical Prostatectomy in Patients with Biopsy Gleason Score 8–10

**DOI:** 10.3390/jcm8040438

**Published:** 2019-03-30

**Authors:** Doo Yong Chung, Jong Soo Lee, Hyeok Jun Goh, Dong Hoon Koh, Min Seok Kim, Won Sik Jang, Young Deuk Choi

**Affiliations:** 1Department of Urology, Inha University School of Medicine, 366 Seohae-daero, Jung-gu, Incheon 22332, Korea; wjdendyd@gmail.com; 2Department of Urology, Urological Science Institute, Yonsei University College of Medicine, 50-1 Yonsei-ro, Seodaemun-gu, Seoul 120-752, Korea; JS1129@yuhs.ac (J.S.L.); JOON8301@yuhs.ac (H.J.G.); DHKOH@yuhs.ac (D.H.K.); KMSVVVV4928@yuhs.ac (M.S.K.); SINDAKJANG@yuhs.ac (W.S.J.)

**Keywords:** prostatic neoplasms, magnetic resonance imaging, gleason score

## Abstract

Gleason score (GS) 8–10 is associated with adverse outcomes in prostate cancer (PCa). However, biopsy GS (bGS) may be upgraded or downgraded post-radical prostatectomy (RP). We aimed to investigate predictive factors and oncologic outcomes of downgrade to pathologic GS (pGS) 6–7 after RP in PCa patients with bGSs 8–10. We retrospectively reviewed clinical data of patients with bGS ≥ 8 undergoing RP. pGS downgrade was defined as a pGS ≤ 7 from bGS ≥ 8 post-RP. Univariate and multivariate cox regression analysis, logistic regression analysis, and Kaplan–Meier curves were used to analyze pGS downgrade and biochemical recurrence (BCR). Of 860 patients, 623 and 237 had bGS 8 and bGS ≥ 9, respectively. Post-RP, 332 patients were downgraded to pGS ≤ 7; of these, 284 and 48 had bGS 8 and bGS ≥ 9, respectively. Prostate-specific antigen (PSA) levels; clinical stage; and adverse pathologic features such as extracapsular extension, seminal vesicle invasion and positive surgical margin were significantly different between patients with pGS ≤ 7 and pGS ≥ 8. Furthermore, bGS 8 (odds ratio (OR): 0.349, *p* < 0.001), PSA level < 10 ng/mL (OR: 0.634, *p* = 0.004), and ≤cT3a (OR: 0.400, *p* < 0.001) were identified as significant predictors of pGS downgrade. pGS downgrade was a significant positive predictor of BCR following RP in patients with high bGS (vs. pGS 8, hazard radio (HR): 1.699, *p* < 0.001; vs. pGS ≥ 9, HR: 1.765, *p* < 0.001). In addition, the 5-year BCR-free survival rate in patients with pGS downgrade significantly differed from that in patients with bGS 8 and ≥ 9 (52.9% vs. 40.7%, *p* < 0.001). Among patients with bGS ≥ 8, those with bGS 8, PSA level < 10 ng/mL, and ≤cT3a may achieve pGS downgrade after RP. These patients may have fewer adverse pathologic features and show a favorable prognosis; thus we suggest that active treatment is needed in these patients. In addition, patients with high-grade bGS should be managed aggressively, even if they show pGS downgrade.

## 1. Introduction

Prostate cancer (PCa) is the most common type of newly diagnosed malignancy in men [1], and it accounts for nearly 30% of all diagnosed male cancers [2]. Prostate-specific antigen (PSA) measurements are widely used for PCa screening [3], and patients with elevated PSA levels undergo prostate biopsy for PCa diagnosis [4]. The Gleason score (GS) is the most commonly used grading system for PCa. Gleason score at prostate biopsy (bGS) is one of the important parameters in the evaluation, risk stratification, and selection of treatment modality in PCa patients [5,6]. Gleason score was first introduced in 1974 by Donald Gleason and the Veterans Administration Cooperative Urologic Research Group [7]. GS is continuously updated, and it was first revised in 2005 in the International Society of Urological Pathology (ISUP) conference [8]. In the 2014 ISUP meeting, GS was further revised, and a new 5-step Gleason grading system was introduced [9]. Despite these revisions, bGS ≥8 is still considered as high risk [10].

Biopsy and radical prostatectomy (RP) specimens can be discordant due to sampling errors and the multifocal nature of PCa [11]. Several studies have already reported GS discrepancies between biopsy and RP specimens [12,13]. A GS downgrade may also occur in patients with bGS ≥ 8 [14,15]. However, few studies have predicted the discordance in GS between biopsy and RP specimens in patients with bGS ≥ 8. We believe that predicting pathological downgrade (pGS ≤ 7) in patients with high-grade prostate cancer may be beneficial in terms of oncologic outcomes and provide a reliable parameter for defining the appropriate surgical treatment strategy. Therefore, this study aimed to investigate prognostic factors of GS downgrade from biopsy to RP and to investigate oncologic outcomes after RP in patients with bGS ≥ 8.

## 2. Materials and Methods

### 2.1. Patients and Study Design

We retrospectively retrieved the clinical and pathological data of 5438 PCa patients who underwent RP at our institution between January 2005 and December 2016. Patients who had undergone androgen deprivation therapy or external beam radiation therapy before RP and those with incomplete pathological or follow-up data were excluded. All patients were diagnosed with PCa via transrectal ultrasound (TRUS)-guided 12-core systematic needle biopsy.

The clinical characteristics of these patients including age, body mass index (BMI), preoperative PSA levels, prostate volume measured via TRUS, clinical stage on multiparametric magnetic resonance imaging (mpMRI), GS following prostate biopsy, and pathologic characteristics of RP specimens were obtained through a review of medical records. The mpMRI images included standardized criteria for Likert scoring of multiparametric sequences using a 3.0-T MRI system (Intera Achieva 3.0-T, Phillips Medical System, Best, The Netherlands) [16]. All pathologic diagnoses were performed by expert pathologists. Biopsy specimens obtained from other hospitals were reviewed by our pathologists. The pathologists also reviewed the missing data in pathological reports of patients included in our study and confirmed that there was no problem. The final identification was performed by the most skilled pathologist to reduce possible errors between pathologists. We assigned GS according to the 2005 ISUP Modified Gleason System [8]. Gleason score at prostate biopsy was based on the core with the highest GS. In case of multifocal disease, GS of the RP specimen was assigned based on the nodule with the highest GS. Tertiary Gleason 5 patterns were assigned if the tertiary component comprised <5% of the entire tumor [17]. Gleason score downgrade was defined as a downgrade to GS ≤ 7 in the RP specimen from a bGS ≥ 8, which is classified as high-grade.

Finally, the tumor-node-metastasis (TNM) stage was determined according to the 8th edition of the American Joint Committee on Cancer TNM staging system.

### 2.2. Follow-Up

Follow-up postoperative PSA test was performed monthly for the first 6 months, every 3 months until the second year, and every 6 months thereafter. Biochemical recurrence (BCR) was defined as any two consecutive increases in serum PSA ≥ 0.2 ng/mL following RP. Biochemical recurrence -free survival was defined as the time from RP to BCR [18]. The follow-up period was calculated from the time of RP to the date of the last known contact with the patient.

### 2.3. Statistical Analysis

Univariate and multivariate Cox proportional hazards regression analyses were performed to assess the association between baseline parameters and BCR-free survival. In addition, univariate and multivariate logistic regression were performed to identify significant factors of pathologic downgrade. The Kaplan–Meier method with log-rank tests were performed to estimate and compare oncologic outcomes according to variations in GS from RP.

Significant variables from univariate analysis were included in multivariate analysis. All statistical analyses were performed using SPSS Statistics software, version 23.0 (IBM, Armonk, NY, USA), and *p* < 0.05 was considered statistically significant.

### 2.4. Ethics Approval and Informed Consent

The current research was approved by the Severance hospital institutional review board (approval number 4-2018-1012). Informed consent from the participants was waived by the institutional review board as the current study satisfied all of the following requirements for the waiver of informed consent: the research involved no more than minimal risk to the participants (retrospective data analysis of previously collected medical records).

## 3. Results

### 3.1. Patient and Disease Characteristics

Of the 5438 patients identified, 1126 had bGS ≥ 8. After excluding 266 patients who received neoadjuvant therapy, 860 patients were finally included in the analysis. In total, 623 (72.4%) and 237 (27.6%) patients had bGS 8 and bGS ≥ 9, respectively. In the 860 specimens collected following RP, the GS were 6, 7, 8, and ≥ 9 in 19 (2.2%), 313 (36.4%), 219 (25.5%), and 309 (35.9%) specimens, respectively. Among 313 patients in GS 7, 130 (15.1%) had GS (3 + 4) and 183 (21.2%) had GS (4 + 3). Therefore, 332 (38.6%) patients with pGS ≤ 7 were followed after RP for pGS downgrade; the remaining 528 (61.4%) patients had pGS ≥ 8. Median follow-up period after RP was 51 months (interquartile range, 26–78 months). In addition, patients with tertiary G5 patterns in the total RP specimen were observed 15 in GS7 (3 + 4), 23 in GS7 (4 + 3) and 15 in GS8 (4 + 4).

When dividing the two groups (pGS ≤ 7 vs. pGS > 8) based on pathologic GS, there were significant differences in preoperative PSA, PSA density, clinical stage, and bGS between the two groups. Meanwhile, age, BMI, prostate volume, and follow-up duration did not differ significantly.

After RP, there were statistically significant differences in the pathological features (extracapsular extension (ECE), seminal vesicle invasion (SVI), positive surgical margin (PSM), lymphovascular invasion, and perineural invasion) between the two groups (Table 1).

### 3.2. Preoperative Factors Related to Pathologic Gleason Score (GS) Downgrade (pGS ≤ 7)

In this study, we used univariate and multivariate logistic regression analyses to identify preoperative predictors of pathologic GS downgrade. Preoperative PSA ≥ 10 ng/mL (odds ratio (OR): 0.606, 95% confidence interval (CI): 0.438–0.840, *p* = 0.003), bGS ≥ 9 (OR: 0.303, 95% CI: 0.213–0.432, *p* < 0.001), and ≥cT3b on mpMRI (OR: 0.284, 95% CI: 0.176–0.458, *p* < 0.001) were found to be independent predictors of pathologic GS downgrade at RP in both univariate and multivariate models (Table 2). Also, in relate of pathologic downgrade, the area under the curves of the receiver operating characteristic curve was 0.614 (95% CI: 0.575–0.653) in patients with bGS 8, PSA level < 10 ng/mL, and ≤cT3a.

### 3.3. Supplementary Analysis of 3-2 (646 Cases)

We analyzed the presence of low-grade PCa core in the biopsy specimens in patients with bGS ≥ 8. In total of 646 cases, we were able to analyze all 12 biopsy cores, including the highest GS. Among the positive biopsy cores, the patients containing Low GS (bGS < 8) was 356 cases of them. In our result, presence of low GS core in biopsy was statistically significant with pathologic downgrade. (OR 3.967, 95% Cl: 2.698–5.833, *p* < 0.001). In addition, we analyzed according to the lowest GS in biopsy. The cases including bGS6 were OR 4.926, 95% CI: 3.119–7.779, *p* < 0.001. The cases including bGS7 (3 + 4) were OR 4.299, 95% CI: 2.566–7.204, *p* < 0.001. And the cases including bGS7 (4 + 3) were OR 2.259, 95% CI: 0.176–0.458, *p* = 0.006.

### 3.4. Oncologic Outcomes and Prognostic Factors

During the follow-up period, 502 cases (58.4%) of BCR were noted. Univariate and multivariate Cox regression analyses were performed with each clinical parameter for BCR. In these analyses, preoperative PSA (hazard ratio [HR]: 1.005, *p* = 0.003), PSA ≥10 ng/mL (HR: 1.53, *p* < 0.001), bGS ≥9 (HR: 1.233, *p* = 0.042), pGS (pGS ≤7 vs. pGS 8: HR: 1.699, *p* < 0.001 and vs. pGS ≥9: HR: 1.765, *p* < 0.001), SVI (HR: 1.820, *p* < 0.001), and PSM (HR: 1.819, *p* < 0.001) were all independent prognostic factors for BCR (Table 3).

Analysis according to bGS and pGS using Kaplan–Meier curves was also conduced. In the pGS downgrade group, the Kaplan–Meier curve showed significantly better BCR-free survival than that in the other group (log-rank test, *p* < 0.001). The 5-year BCR-free survival rates was higher in the pGS downgrade group than that in the other group (50.9% vs. 23.8%). Also, patients with bGS 8, PSA < 10 ng/mL, and ≤ cT3a showed significantly better 5-year BCR-free survival than the other patients (55.1% vs. 23.4%, *p* < 0.001; Figure 1 and Figure 2).

The patients were then divided into four groups as follows for further analysis: group 1 comprised patients with bGS 8 and pGS < 7; group 2, patients with bGS ≥ 9 and pGS < 7; group 3, patients with bGS ≥ 9 and pGS ≥ 8; and group 4, patients with bGS ≥ 9 and pGS ≥ 8. The 5-year BCR-free survival rates for groups 1, 2, 3, and 4 were 52.9%, 40.7%, 27.2%, and 16.9%, respectively. The hazard ratios of groups 2, 3, and 4 compared to group 1 were 2.109 (*p* = 0.146), 38.643 (*p* < 0.001), and 96.262 (*p* < 0.001), respectively. The BCR-free survival curves of the four groups are shown in Figure 3.

## 4. Discussion

Since the introduction of the Gleason Grading System [7], bGS 8 or higher has been consistently classified as high risk and as a predictor of poor oncologic outcome. Patients with bGS 8 are classified to have high-grade prostate cancer (poorly differentiated prostate cancer). Since the 2005 update in the Gleason grading system [8], the correlation between bGS and RP GS has improved [19,20,21]. Major changes in the Gleason system have also been made during the 2014 ISUP [9], and this further improves the association between bGS and pGS [22]. However, some cases of discordant bGS and pGS are still reported. Among patients with PCa, those with GS > 8, PSA > 20 ng/mL, or cT3a are considered to be high risk. Although RP is a treatment option in these patients, combined EBRTx and ADT are proposed as the first choice of treatment in the National Comprehensive Cancer Network (NCCN) guideline [10]. The European Association of Urology (EAU) guideline also recommends RP for selected patients [4], and both the EAU and NCCN guidelines recommend RP in select high-risk patients. Therefore, in this study, we investigated the favorable indicators for surgical treatment in patients with high-risk PCa.

pGS downgrade is a well-known favorable prognostic factor, and Donohue et al. [14] reported that patients with high bGS who have pGS downgrade have good prognosis. In their study, cT1c and bGS 8 were identified to be predictors of pGS downgrade, which is similar to our results. However, there were differences about the influence of PSA and clinical stage between the study of Donohue et al. and our study. In our study, PSA was set as a criterion for easy patient and used a PSA value of 10 ng/mL as cut-off for low-risk malignancy based on the current standard [10]. Thus, PSA <10 ng/mL was evaluated as a predictor for downgrade in patients with high-grade PCa on biopsy findings. In addition, recent developments in mpMRI have improved the image evaluation techniques for ECE and SVI [23,24]. The clinical stage was evaluated using 3T MRI, which enabled us to distinguish between T3a and T3b. Thus, the probability of discordance (i.e., pathologic downgrade) in patients with T3b or higher was minimized. In addition, we found that bGS8, PSA < 10 ng/mL, and ≤ cT3a were significantly correlated with BCR in patients with high bGS. These factors were also predictors of pathologic downgrade. The Kaplan–Meier curves of the patients with pathologic downgrade and those with bGS8, PSA < 10 ng/mL, and ≤ cT3a were similar in this study.

Collectively, these results indicate that patients with high-grade PCa who have PSA < 10 ng/mL, bGS 8, and ≤ cT3a may have favorable oncological outcomes via RP. This information would be valuable in providing an indication for surgical treatment in patients with high-grade PCa. In addition, we believe that bGS remains a crucial factor for determining the appropriate treatment and follow-up strategy for high-grade PCa.

The oncologic outcome has been reported to differ according to bGS among those with similar pGS [11,22]. In our study, we also found that BCR-free survival decreased as bGS increased among those with similar pGS. This suggests that although pGS downgrades may be observed in patients with high-grade PCa on biopsy, these patients require a more aggressive management after RP than those with low-grade PCa.

Our study has several limitations. First, this was a retrospective review of data from patients treated at a single institution. Thus, our results are subject to selection bias, limiting generalizability. Therefore, multi-center, prospective studies are needed. Second, mpMRI findings have been recently read based on the Prostate Imaging Reporting and Data System (PI-RADS) [25,26]. However, until 2009 we did not include diffusion-weighted imaging and the apparent-diffusion coefficient in the mpMRI protocol. Therefore, it was difficult to perform a complete PI-RADS scoring. Recently, PI-RADS scoring has been reported to be associated with GS [27,28], and this requires further analysis. Third, given the referral nature of our tertiary center, many men underwent biopsy at outside hospitals. While all biopsy slides were re-examined by a single experienced group of urological pathologists at our institute, biopsy data were lacking in some cases, such as the number of positive cores, percentage of cancer per core, and perineural invasion. In our study, the presence of low GS (bGS <8) core in positive biopsy cores and pathologic downgrade were statistically significant. Unfortunately, as mentioned earlier, biopsy data were lacking in some cases. Therefore, it was difficult to analysis the average cancer portion for all biopsy cores. We think that this analysis may be a good study to reduce the discordance of biopsy and final GS. However, as a tertiary center, we were able to accommodate many patients with high-grade PCa. Therefore, a large number of patients were included in this study. In addition, in the era of mpMRI, the index core via mpMRI fusion biopsy could be one factor that could reduce GS discrepancies between biopsy and RP specimen. Unfortunately, this was not included in our current study. Therefore, we believe that further research on this content is necessary.

Despite these limitations, our study is still valuable in that it demonstrates that bGS 8, PSA < 10 ng/mL, and < cT3b were independent predictors of downgrading GS after RP in patients with high-grade PCa. In addition, these factors were determined to be predictors of favorable oncological outcome in these patients. Our findings provide a guide for predicting disease prognosis and subsequent treatment planning in patients with high-grade PCa.

## 5. Conclusions

Among patients with high-risk PCa (i.e., those with bGS ≥ 8), those with bGS 8, PSA < 10 ng/mL, and without SVI on mpMRI may have pGS downgrade and favorable oncologic outcomes. These patients have fewer adverse pathologic features and can thus benefit from RP in terms of oncologic prognosis. Therefore, we suggest active treatment such as RP in these patients.

## Figures and Tables

**Figure 1 jcm-08-00438-f001:**
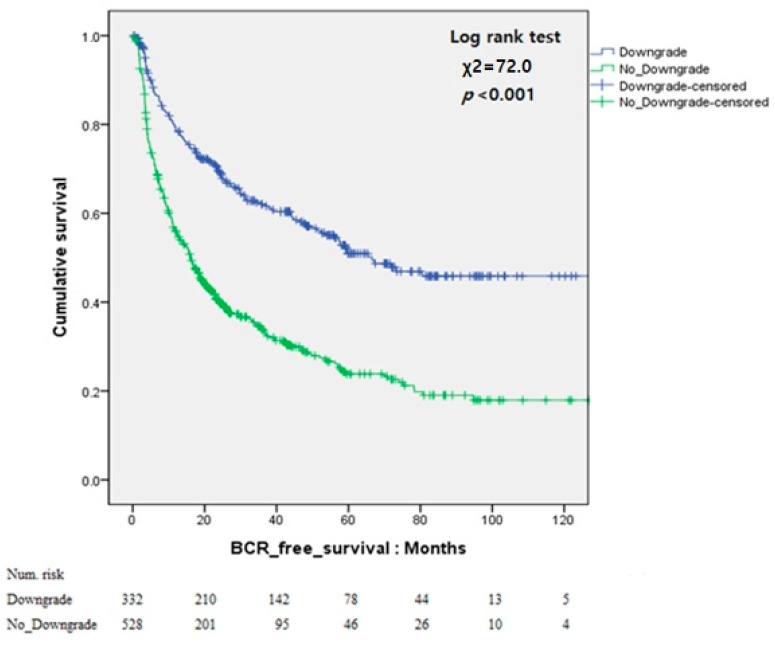
Kaplan–Meier curves for biochemical recurrence (BCR)-free survival according to pathologic Gleason score downgrade.

**Figure 2 jcm-08-00438-f002:**
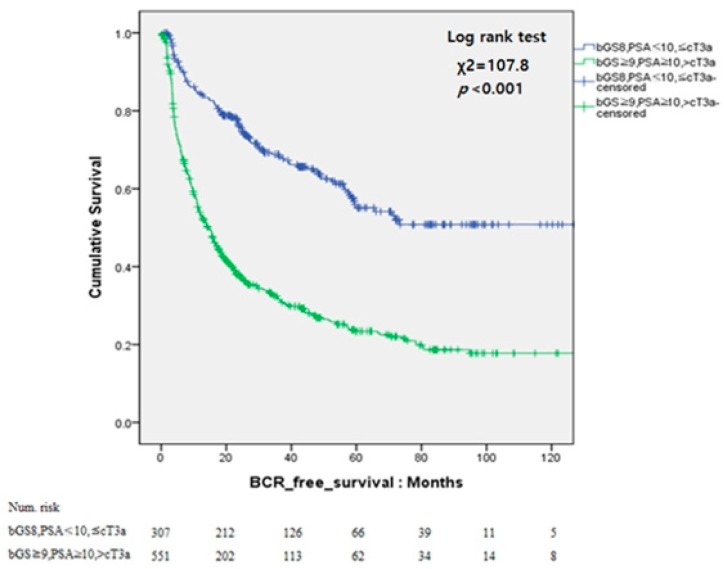
Kaplan–Meier curves for biochemical recurrence (BCR)-free survival according to bGS 8, PSA < 10 ng/mL, and ≤ cT3a.

**Figure 3 jcm-08-00438-f003:**
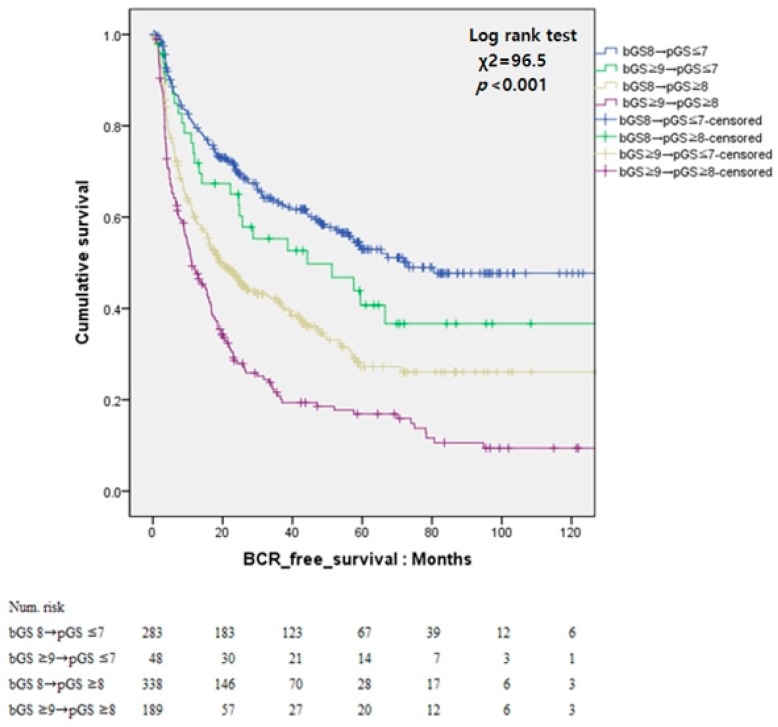
Kaplan–Meier curves for biochemical recurrence (BCR)-free survival according to bGS and pGS.

**Table 1 jcm-08-00438-t001:** Baseline patient characteristics.

Variable	Total		No_Downgrade		Downgrade		*P* value
	*N* = 860		*N* = 528 (61.4%)		*N* = 332 (38.6%)		
	Median	IQR	Median	IQR	Median	IQR	
Age, year	67	62–71	67	62–71	66	62–70	0.578
BMI, kg/m^2^	24.09	22.38–25.97	24.05	22.32–25.90	24.22	22.50–26.00	0.608
PSA level, ng/mL	10.59	6.58–19.39	12.03	7.09–22.72	8.60	5.99–14.74	<0.001
PSA group, ng/mL	N	%	N	%	N	%	<0.001
<10	445	51.7	236	44.7	209	63.0	
≥10	415	48.3	292	55.3	123	37.0	
PSA density, ng/mL^2^	0.33	0.20–0.62	0.38	0.22–0.75	0.28	0.18–0.50	<0.001
Prostate volume, mL	32.0	26.0–40.5	32.6	26.0–40.9	31.1	25.8–40.0	0.367
Clinical stage according to MRI	N	%	N	%	N	%	<0.001
cT2	402	46.7	213	40.3	35	56.9	
cT3a	292	34.0	176	33.3	116	34.9	
≥cT3b	166	19.3	139	26.3	27	8.1	
Gleason score according to Biopsy	N	%	N	%	N	%	<0.001
8	623	72.4	339	64.2	284	85.5	
≥9	237	27.6	189	35.8	48	14.5	
FU duration after RP, months	51	26–78	44	25–71	59	32–83	0.317
**Pathologic Features after Radical Prostatectomy**							
**Gleason Score according** **to Radical Prostatectomy**	**N**	**%**	**N**	**%**	**N**	**%**	
6	19	2.2	-		19	5.7	
7 (3 + 4)	130	15.1	-		130	39.2	
7 (4 + 3)	183	21.2	-		183	55.1	
8	219	25.5	219	41.5	-		
≥9	309	35.9	309	58.6	-		
**Pathologic T Stage according** **to Radical Prostatectomy**	**N**	**%**	**N**	**%**	**N**	**%**	**<0.001**
T2	260	30.2	118	22.3	142	42.8	
T3a	348	40.5	208	39.4	140	42.2	
≥T3b	252	29.3	202	38.3	50	15.1	
ECE	570	66.7	394	74.6	180	54.2	<0.001
SVI	214	24.9	164	31.1	50	15.1	<0.001
PSM	452	52.6	300	56.8	152	45.8	0.002
LVI	11	12.7	84	15.9	25	7.5	<0.001
PNI	629	73.1	405	76.7	224	67.5	0.003
HGPIN	371	43.1	204	38.6	167	50.3	0.001
BCR	502	58.4	360	68.2	142	42.8	<0.001

IQR, interquartile range; BMI, body mass index; PSA, prostate-specific antigen; MRI, magnetic resonance imaging; RP, radical prostatectomy; ECE, extracapsular extension; SVI, seminal vesicle invasion; PSM, positive surgical margin; LVI, lymphovascular invasion; PNI, perineural invasion; HGPIN, high-grade prostatic intraepithelial neoplasia; BCR, biochemical recurrence.

**Table 2 jcm-08-00438-t002:** Univariate and multivariate analyses of factors associated with pathologic Gleason score downgrade in patients with biopsy Gleason scores 8–10.

Variable	Univariate		Multivariate	
	OR (95% CI)	*p* Value	OR (95% CI)	*p* Value
Age, year	0.995 (0.975–1.015)	0.592		
BMI, kg/m^2^	1.013 (0.964–1.064)	0.607		
Prostate volume, mL	0.996 (0.986–1.005)	0.367		
PSA level, ng/mL				
<10	1 (Ref)		1 (Ref)	
≥10	0.476 (0.359–0.630)	<0.001	0.606 (0.438–0.840)	0.003
PSA density	0.679 (0.540–0.854)	0.001	0.973 (0.799–1.185)	0.783
Gleason score				
8	1 (Ref)		1 (Ref)	
≥9	0.303 (0.213–0.432)	<0.001	0.342 (0.238–0.493)	<0.001
Clinical stage according to MRI				
cT2	1 (Ref)		1 (Ref)	
cT3a	0.743 (0.547–1.008)	0.056	0.816 (0.592–1.123)	0.211
≥cT3b	0.219 (0.139–0.346)	<0.001	0.284 (0.176–0.458)	<0.001

OR, odds ratio; CI, confidence interval; Ref, reference value; BMI, body mass index; PSA, prostate-specific antigen; MRI, magnetic resonance imaging.

**Table 3 jcm-08-00438-t003:** Univariate and multivariate analyses of factors associated with biochemical recurrence.

Variable	Univariate		Multivariate	
	HR (95% CI)	*p* Value	HR (95% CI)	*p* Value
Age, year	1.005 (0.992–1.018)	0.469		
BMI, kg/m^2^	1.014 (0.982–1.046)	0.399		
Prostate volume, ml	1.003 (0.997–1.008)	0.363		
PSA, ng/ml	1.006 (1.005–1.008)	<0.001	1.005 (1.002–1.009)	0.003
<10	1 (Ref)		1 (Ref)	
≥10	2.341 (1.956–2.803)	<0.001	1.539 (1.260–1.880)	<0.001
PSA density, ng/mL^2^	1.186 (1.127–1.249)	<0.001	0.888 (0.780–1.011)	0.073
Biopsy Gleason score (bGS)				
8	1 (Ref)		1 (Ref)	
≥9	1.784 (1.485–2.143)	<0.001	1.233 (1.008–1.508)	0.042
Pathologic Gleason score (pGS)				
≤7 (pGS_downgrade)	1 (Ref)		1 (Ref)	
8	1.735 (1.366–2.204)	<0.001	1.699 (1.328–2.175)	<0.001
≥9	2.810 (2.272–3.474)	<0.001	1.765 (1.396–2.231)	<0.001
Pathologic T stage				
≤T2	1 (Ref)		1 (Ref)	
≥T3	5.044 (3.910–6.508)	<0.001	1.409 (0.788–2.520)	0.247
ECE				
No	1 (Ref)		1 (Ref)	
Yes	2.641 (2.122–3.286)	<0.001	1.016 (0.594–1.737)	0.955
SVI				
No	1 (Ref)		1 (Ref)	
Yes	3.139 (2.598–3.792)	<0.001	1.820 (1.473–2.249)	<0.001
PSM				
No	1 (Ref)		1 (Ref)	
Yes	2.775 (2.298–3.351)	<0.001	1.819 (1.476–2.242)	<0.001
LVI				
No	1 (Ref)		1 (Ref)	
Yes	2.069 (1.639–2.612)	<0.001	1.260 (0.987–1.608)	0.063
PNI				
No	1 (Ref)		1 (Ref)	
Yes	2.013 (1.612–2.512)	<0.001	1.244 (0.979–1.580)	0.074
HGPIN				
No	1 (Ref)			
Yes	0.870 (0.728–1.040)	0.126		

HR, hazard ratio; CI, confidence interval; Ref, reference value; BMI, body mass index; PSA, prostate-specific antigen; ECE, extracapsular extension; SVI, seminal vesicle invasion; PSM, positive surgical margins; LVI, lymphovascular invasion; PNI, perineural invasion; HGPIN, high-grade prostatic intraepithelial neoplasia; BCR, biochemical recurrence.

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
