# Peer review of "Predictive Factors and Oncologic Outcome of Downgrade to Pathologic Gleason Score 6–7 after Radical Prostatectomy in Patients with Biopsy Gleason Score 8–10"

_jcm, 2019, doi:10.3390/jcm8040438_

Round 1
Reviewer 1 Report
In this impressively large study, Chung et al. described a cohort of 860 patients for which Gleason grade at biopsy, Gleason grade of the subsequent prostatectomy and follow-up data are available. The cohort exclusively includes patients that had a Gleason 8-9 on their diagnostic biopsies. The authors show that patients of which the Gleason grade was downgraded from biopsy to the prostatectomy sample, as well as patients with low PSA and low clinical stage had a particularly favourableprognosis among these high-risk patients. The strength of the study certainly is the very large patient cohort with a comparable clinical situation. Therefore, the dataset should in principle be published. There are, however, two major issues that should be addressed.
1. It is unclear, how the biopsy Gleason grade was defined. Was the Gleason grade used for this study the worst Gleason grade found among multiple cancer positive biopsies? Or was the average Gleason grade ofall biopsies used. This reviewer strongly assumes, that a low average Gleason grade (for example in patients with one biopsy 4+5 and several others with 3+4, 4+3 or even 3+3) would be an alternative explanation for the good prognosis of patients with a Gleason downgrade. The authors should clarify this issue and provide data on the prognostic impact of the average findings in the biopsies.
2. In their discussion, the authors should also focus on the issue of interobserver variability for the Gleason grading, which - according to the literature - is at least as important for Gleason up- and downgrading as sampling issues. This reviewer assumes that much of the discrepancies between biopsy and prostatectomy Gleason are due to interobserver variability. If this was true, many cases where two observers disagree would represent borderline cases (between Gleason 8-10 and Gleason 7). The good prognosis of downgraded prostate cancers could thus be explained by the borderline nature of the Gleason.
The main weakness of the paper is a relative lack of novelty because the prognostic impact of Gleason downgrading, clinical stage and PSA levels are already known. The authors could add substantial value to their manuscript if they utilized their data to shed more light on certain controversial issues in the Gleason grading, especially on the issue, whether the worst or the average biopsy result should be used to categorize patients in a preoperative setting. In this regard the authors might have relevant data.
Minor issue: How did the authors deal with minor quantities of Gleason 5 in downgraded prostate cancer prostatectomy specimens? How did they define tertiary Gleason 5 patterns? How often Gleason 5 patterns, described in the biopsy could not be reproduced in the prostatectomy specimen.
Author Response
Thank you for your thoroughly reviewing our manuscript (jcm-451461) entitled “Predictive factors and oncologic outcome of downgrade to pathologic Gleason score 6-7 after radical prostatectomy in patients with biopsy Gleason score 8-10” Also, we are grateful for the chance to revise our manuscript. Our manuscript has been carefully revised according to the reviewers’ comments. Please find our responses to the reviewer’ comments beginning on the next page.
We hope that our revised paper is acceptable for publication in Journal of Clinical Medicine, and we look forward to receiving your final decision.
Thanks, again.
Sincerely,
Comment
In this impressively large study, Chung et al. described a cohort of 860 patients for which Gleason grade at biopsy, Gleason grade of the subsequent prostatectomy and follow-up data are available. The cohort exclusively includes patients that had a Gleason 8-9 on their diagnostic biopsies. The authors show that patients of which the Gleason grade was downgraded from biopsy to the prostatectomy sample, as well as patients with low PSA and low clinical stage had a particularly favourableprognosis among these high-risk patients. The strength of the study certainly is the very large patient cohort with a comparable clinical situation. Therefore, the dataset should in principle be published. There are, however, two major issues that should be addressed.
Answer
Thank you for your comment. We have modified this paper according to your feedback. Due to your comment, we think that this paper has become a more valuable literature.
Comment
1. It is unclear, how the biopsy Gleason grade was defined. Was the Gleason grade used for this study the worst Gleason grade found among multiple cancer positive biopsies? Or was the average Gleason grade ofall biopsies used. This reviewer strongly assumes, that a low average Gleason grade (for example in patients with one biopsy 4+5 and several others with 3+4, 4+3 or even 3+3) would be an alternative explanation for the good prognosis of patients with a Gleason downgrade. The authors should clarify this issue and provide data on the prognostic impact of the average findings in the biopsies.
Answer
Thank you for your comment. In this study, we basically performed GS based on 2005 ISUP. Biopsy GS was based on the core with the highest GS. In multifocal disease, RP GS was assigned to the nodule with the highest GS. This refers to the previous papers. A tendency to assign higher Gleason scores for needle biopsy specimens has been observed during the past decades. The 2005 ISUP revision has undoubtedly contributed to this. But the more frequent use of higher Gleason scores for biopsy is associated with improved agreement between needle biopsy and RP remains unclear. (Location of the content: 73-79)
So, based on your opinion, we added the difference in GS downgrade between with and without biopsy GS low core. Unfortunately, as mentioned earlier, we lacked some biopsy results. In total of 646 cases, we were able to analyze about all 12 biopsy cores including the highest GS. In result, presence of low grade (<bGS 8) core in positive biopsy cores and pathologic downgrade were statistically significant. So we have added a supplementary analysis type in the Results section. (Location of the content: 136-144) Because there were some deficiencies, it was difficult to analysis the average cancer portion for all biopsy cores. We think that this analysis may be a good study to reduce the discordance of biopsy and final GS. This was added as a limitation in discussion section. (Location of the content: 235-239)
Comment
2. In their discussion, the authors should also focus on the issue of interobserver variability for the Gleason grading, which - according to the literature - is at least as important for Gleason up- and downgrading as sampling issues. This reviewer assumes that much of the discrepancies between biopsy and prostatectomy Gleason are due to interobserver variability. If this was true, many cases where two observers disagree would represent borderline cases (between Gleason 8-10 and Gleason 7). The good prognosis of downgraded prostate cancers could thus be explained by the borderline nature of the Gleason.
Answer
Thank you for your comment. We are conducting the final confirm by the most skilled pathologist to prevent errors that may occur between pathologists. (Location of the content: 73-74)
However, as you said, we would not have prevented all interobserver errors. Thank you.
Comment
The main weakness of the paper is a relative lack of novelty because the prognostic impact of Gleason downgrading, clinical stage and PSA levels are already known. The authors could add substantial value to their manuscript if they utilized their data to shed more light on certain controversial issues in the Gleason grading, especially on the issue, whether the worst or the average biopsy result should be used to categorize patients in a preoperative setting. In this regard the authors might have relevant data.
Answer
Thank you very much for your comments on how to improve our paper. We have modified it on the basis of what you have said.
Comment
Minor issue: How did the authors deal with minor quantities of Gleason 5 in downgraded prostate cancer prostatectomy specimens? How did they define tertiary Gleason 5 patterns? How often Gleason 5 patterns, described in the biopsy could not be reproduced in the prostatectomy specimen.
Answer
Thank you for your comment. Tertiary Gleason 5 patterns was assigned if the tertiary component comprised less than 5% of the entire tumor. In addition, patients with tertiary G5 in the total RP specimen were observed 15 in GS7(3+4), 23 in GS7 (4+3) and 15 in G8 (4+4). (Location of the content: 106-107)
Reviewer 2 ReportThe aim of this study is to evaluate predictive factors and oncologic outcomes of downgrading to pathological GS (pGS) after radical prostatectomy. The topic is actual and could have an important role in the disease management of Pca, even if there are some manuscript’s aspects that could be reviewed.
The abstract is too concise; the reader not understand the topic of the study and the endpoints are not clear. There are reported in detail the results of the study that is not necessary.
Which is the definition of downgrading to pathological GS? This could be reported clearly in the material and methods.
In the era of mpMRI why did not the Author mention fusion biopsies? Fusion biopsies could reduce the GS downgrading after RP due to target biopsies performed into the index lesions that are more probably clinical significant Pca. In my opinion this is one of the important limitation of the study, that is not reported.
In the introduction it is reported the ISUP grade group classification that is introduced since 2004. Why the Author not used this classification in the study that divide GS 7 (3+4) and GS 7 (4+3) in two different groups?
The statistical analysis is clear and the results well reported. Tables and figures help the reader to interpret the results obtained.
In conclusion in my opinion the study has some critical gaps that needs to be filled, even if the aim of the study and the results obtained are interesting and could have an important role in the disease management of Pca.
Author Response
Thank you for your thoroughly reviewing our manuscript (jcm-451461) entitled “Predictive factors and oncologic outcome of downgrade to pathologic Gleason score 6-7 after radical prostatectomy in patients with biopsy Gleason score 8-10” Also, we are grateful for the chance to revise our manuscript. Our manuscript has been carefully revised according to the reviewers’ comments. Please find our responses to the reviewer’ comments beginning on the next page.
We hope that our revised paper is acceptable for publication in Journal of Clinical Medicine, and we look forward to receiving your final decision.
Thanks, again.
Sincerely,
Comment
The aim of this study is to evaluate predictive factors and oncologic outcomes of downgrading to pathological GS (pGS) after radical prostatectomy. The topic is actual and could have an important role in the disease management of Pca, even if there are some manuscript’s aspects that could be reviewed.
The abstract is too concise; the reader not understand the topic of the study and the endpoints are not clear. There are reported in detail the results of the study that is not necessary.
Answer
Thank you for your comment. We added the definition of downgrading and modified abstract to focus on downgrading.
(Location of the content: 15-34)
Comment
Which is the definition of downgrading to pathological GS? This could be reported clearly in the material and methods.
Answer
Thank you for your comment. We have included definitions in the material and methods section.
(Location of the content: 73-79)
Comment
In the era of mpMRI why did not the Author mention fusion biopsies? Fusion biopsies could reduce the GS downgrading after RP due to target biopsies performed into the index lesions that are more probably clinical significant Pca. In my opinion this is one of the important limitation of the study, that is not reported.
Answer
Thank you for your comment. It was not easy to implement prebiopsy mpMRI in South Korea medical system as a matter of insurance. Fortunately, recently (about 2years ago), with the consent of the patient, the mpMRI fusion biopsy was available. We are collecting these cases. Therefore, it was difficult to include the findings of the mpMRI fusion biopsy in this study. According to your opinion, the study of the GS correlation of biopsy and specimen in mpMRI fusion biopsy patients may be a crucial additional study point. We will add it to the discussion. Thank you very much.
(Location of the content: 241-243)
Comment
In the introduction it is reported the ISUP grade group classification that is introduced since 2004. Why the Author not used this classification in the study that divide GS 7 (3+4) and GS 7 (4+3) in two different groups?
Answer
Thank you for your comment. Based on GS8 or more, which are classified as high grade, we defined the downgrade as GS7 or less (GS6, GS 3+4 and GS 4+3). For this reason, we did not divide GS7 in detail. Sorry. We have further analyzed GS (3+4) and GS (4+3) separately for GS7. In our study, among 313 patients in GS7, GS(3+4) was 130 and GS(4+3) was 183. We added it in table 1. Thank you.
(Location of the content: 103)
Comment
The statistical analysis is clear and the results well reported. Tables and figures help the reader to interpret the results obtained.
In conclusion in my opinion the study has some critical gaps that needs to be filled, even if the aim of the study and the results obtained are interesting and could have an important role in the disease management of Pca.
Answer
Thank you for your comment. We have modified this paper according to your feedback. Due to your comment, we think that this paper has become a more valuable literature.
Round 2
Reviewer 2 Report
Dear Authors,
Thank you for the revisions. In my opinion the topic of the study is interesting and could have an important role in the disease management of Pca. I think that your revisions could improve the quality of the study.